# NH₃ spatio-temporal variability over Paris, Mexico and Toronto and its link to PM2.5 during pollution events

Camille Viatte[1], Rimal Abeed[1], Shoma Yamanouchi[2,3], William Porter[4], Sarah Safieddine[1], Martin Van Damme[5,6], Lieven Clarisse[4], Beatriz Herrera[2,7], Michel Grutter[7], Pierre-Francois Coheur[4], Kimberly Strong[2], and Cathy Clerbaux[1,5].

[1]LATMOS/IPSL, Sorbonne Université, UVSQ, CNRS, 75252 Paris Cedex 05, France;

[2]Department of Physics, University of Toronto, Toronto, ON M5S 1A7, Canada;

[3]Department of Civil and Mineral Engineering, University of Toronto, Toronto ON M5S 1A4, Canada;

[4]Department of Environmental Sciences, University of California, Riverside, CA 92521, USA;

[5]Université libre de Bruxelles (ULB), Spectroscopy, Quantum Chemistry and Atmospheric Remote Sensing (SQUARES), Brussels 1050, Belgium;

[6]BIRA-IASB - Belgian Institute for Space Aeronomy, Brussels 1180, Belgium;

[7]Instituto de Ciencias de la Atmósfera y Cambio Climático, Universidad Nacional Autónoma de México, Mexico City, 04510, Mexico;

*Correspondence:* Camille Viatte (camille.viatte@latmos.ipsl.fr)

Abstract
Megacities can experience high levels of fine particulate matter ($PM_{2.5}$) pollution linked to ammonia
($NH_3$) mainly emitted from agricultural activities. Here, we investigate such pollution in the cities of
Paris, Mexico and Toronto, each of which have distinct emission sources, agricultural regulations, and
topography. Ten years of measurements from the Infrared Atmospheric Sounding Interferometer
(IASI) are used to assess the spatio-temporal $NH_3$ variability over and around the three cities.
In Europe and North America, we determine that temperature is associated with the increase in $NH_3$
atmospheric concentrations with coefficient of determination ($r^2$) of 0.8 over agricultural areas. The
variety of the $NH_3$ sources (industry and agricultural) and the weaker temperature seasonal cycle in
southern North America induce a lower correlation factor ($r^2 = 0.5$). The three regions are subject to
long range transport of $NH_3$, as shown using HYSPLIT cluster back-trajectories. The highest $NH_3$
concentrations measured at the city scales are associated with air masses coming from the surrounding
and north-northeast regions of Paris, the south-southwest areas of Toronto, and the
southeast/southwest zones of Mexico City.
Using $NH_3$ and $PM_{2.5}$ measurements derived from IASI and surface observations from 2008 to 2017,
annually frequent pollution events are identified in the three cities. Wind roses reveal statistical
patterns during these pollution events with dominant northeast-southwest directions in Paris and
Mexico City, and the transboundary transport of pollutants from the United-States in Toronto. To
check how well chemistry transport models perform during pollution events, we evaluate simulations
made using the GEOS-Chem model for March 2011. In these simulations we find that $NH_3$
concentrations are overall underestimated, though day-to-day variability is well represented. $PM_{2.5}$ is
generally underestimated over Paris and Mexico, but overestimated over Toronto.

1. Introduction

Paris, Toronto, and Mexico City are cities with over 2 million inhabitants. When their larger metropolitan regions are included, their populations are 10.5 million for Paris (the most populous area in the European Union), 6.5 million for Toronto (the fourth most populous city in North America) and 9.2 million for Mexico City (most populous city in North America). These cities typically experience strong particulate matter (PM) pollution episodes. Exposure to such particles is harmful to humans and can lead to cardiovascular and respiratory diseases [Murray et al., 2020].

A large proportion of the particles' composition is ammonium sulfate and nitrate, which are formed from ammonia ($NH_3$) [Behera et al., 2013] released in the atmosphere from e.g., fertilizer spreading practices and both transported to cities, reducing the quality of urban air [Pope et al., 2009]. The agricultural sector represents 94%, 90%, and 94% of total $NH_3$ emissions in France [CITEPA, 2018], Canada [ECCC, 2017] and Mexico [INECC and SEMARNAT, 2018], respectively. $NH_3$ is the most poorly understood precursor of $PM_{2.5}$ (PM with a diameter less than 2.5 μm), primarily because measurements are difficult [von Bobrutzki et al., 2010], sparse, and due to low ambient $NH_3$ concentrations and episodic emissions. Worldwide, only five countries (United States, China, Netherlands, United Kingdom, and Canada) have included routine measurements of $NH_3$ concentrations in their air quality monitoring networks [Nair and Yu, 2020].

$NH_3$ emissions are associated with very high uncertainties in all inventories (186% to 294% uncertainties in EDGAR [McDuffie et al., 2020; Van Damme et al., 2018]) due to uncertainties in the reporting of agricultural statistics and emission factors that depend on individual agricultural practices, biological processes, and environmental conditions [Paulot et al., 2014], as well as political disturbances and land-use change [Abeed et al., 2021]. The evaporation of $NH_3$ in the atmosphere, as well as its transformation into particulate matter, is highly dependent on the thermodynamic conditions of the atmosphere [Sutton et al., 2013]. All these parameters account for the complexity of reproducing $NH_3$ concentrations in atmospheric models, predicting the associated $PM_{2.5}$ pollution, and, ultimately, implementing relevant regulations to reduce its emissions.

Given the crucial role that $NH_3$ plays in environmental and public health problems, reducing its emissions will therefore be a major challenge. However, $NH_3$ concentrations are increasing in many countries: France, Canada and Mexico reported increases of 24 ± 11%, 16.4 ± 8.6%, and 8.4 ± 5.2 % between 2008 and 2018 respectively [Van Damme et al., 2021]. These trends are likely explained by increasing emissions, partly due to increased temperature (Europe) and biomass burning (Canada). However, decreasing concentrations of nitrogen and sulfur oxides e.g. in Europe and China also increase the ammonia atmospheric lifetime and plays a role in the reported upward trends.

In Paris, $PM_{2.5}$ are composed of organic matter (38–47 %), nitrate (17–22 %), non-sea-salt sulfate (13–16 %), ammonium (10–12 %), and to a minor extend with elemental carbon, mineral dust (2–5 %) and sea salt [Bressi et al., 2013]. In springtime, it has been shown that $NH_3$ plays a significant role in $PM_{2.5}$ pollution episodes [Viatte et al., 2021] but long-term observations are needed to properly evaluate the impact of $NH_3$ to $PM_{2.5}$ formation.

In Toronto, secondary nitrate formed from nitric acids ($NO_x$) and $NH_3$ account for 36% of the $PM_{2.5}$ sources [Lee et al., 2003] and ammonium nitrate and sulfate accounted for 20-30% of annual $PM_{2.5}$ mass over the 14-year period between 2006 and 2014 [Jeong et al., 2020]. There is a need for a higher number of surface observations to evaluate the $NH_3$-$PM_{2.5}$ relationship and its evolution over time [Larios et al., 2018].

In Mexico, PM$_{2.5}$ concentrations often exceed the national standard of 41 μg/m$^3$ for the 24-hour mean
[NOM-025-SSA1-2021, 2021]. Secondary inorganic aerosols account for 30% of the chemical
composition of PM$_{2.5}$, which are dominated by ammonium sulfate with an average of 14% [Vega et al.,
2010]. A better understanding of the particulate pollutants processes in Mexico is still needed [Ojeda-
Castillo et al., 2019].
To assess the role of NH$_3$ in the formation of particulate matter, the AmmonAQ (Ammonia air quality)
project was designed to quantify NH$_3$ spatio-temporal variabilities in regional domains around these
three cities. The main objective of this project is to determine the impact of intensive agricultural
practices on NH$_3$ and urban air quality, with a focus on Paris, Toronto and Mexico as benchmark case
studies. A schematic representation of the AmmonAQ project and the domains of study are shown in
Figure 1. The so-called "Europe", "North America", and "southern North America" domains represent
the extended area with NH$_3$ sources that can impact on the Paris, Toronto, and Mexico cities air
composition. The three cities are investigated with the use of different datasets: satellite
measurements and model simulation data, and surface measurements when available (see section 2).
These cities have been chosen as the focus of this study because of the availability of NH$_3$ and PM$_{2.5}$
measurements. These three cities differ in terms of:

1. The regulation of NH$_3$ emissions: French policies aim to reduce NH$_3$ emissions by 13% in 2030
   relative to 2005 [CEIP, 2016] following EU ratification of the Gothenburg Protocol in 2017,
   whereas in Canada and Mexico there are no federal regulations for NH$_3$ emissions yet [Bittman
   et al., 2017];
2. Agricultural practices affecting NH$_3$ emissions differ in each region as farmers depend on
   meteorological conditions for fertilizer use;
3. Meteorological/climate conditions are very different in each of the regions: drier winter and
   wetter summer in Toronto compared to Paris, and weak winds and strong temperature
   inversions in Mexico-city. This influences the NH$_3$ lifetime and chemistry leading to the
   formation of PM$_{2.5}$;
4. Topography: Toronto is adjacent to Lake Ontario, Paris is inland, and Mexico-city is a basin
   surrounded by mountains. This will impact the trajectories of air masses.

2. Methodology
2.1. NH$_3$ observations derived from IASI
The Infrared Atmospheric Sounding Interferometer (IASI) was launched onboard the Metop-A/B/C
satellites in 2006, 2012, and 2018, respectively [Clerbaux et al., 2009]. IASI provides twice daily total
column measurements of NH$_3$ globally at 9:30 and 21:30 local solar time. With its polar orbit and a
swath of 2400 km, IASI pixel size is 12 km in diameter at nadir. In this work, we use version 3 of the
ANNI-NH$_3$ product [Van Damme et al., 2021; Guo et al., 2021] from IASI Metop-A/B morning
overpasses over the period 2008 to 2017 gridded a spatial resolution of 0.25° x 0.25°. The detection
limit depends both on the atmospheric state (mainly thermal contrast and NH$_3$ abundance) and the
instrument characteristics. For IASI, the minimum detection limit is found to be 4-6x10$^{15}$
molecules/cm$^2$ [Clarisse et al., 2010].
2.2. PM$_{2.5}$ dataset derived from surface network measurements
To study local scale PM$_{2.5}$ pollution events in the Paris, Toronto, and Mexico cities, PM$_{2.5}$ observations
of surface concentrations from 2008 to 2017 are used.
For Paris, we use hourly observations of $PM_{2.5}$ concentrations derived from fourteen stations of the
Airparif network (https://data-airparif-asso.opendata.arcgis.com/). For Toronto, we analyze hourly
$PM_{2.5}$ observations derived from eleven stations supported by the Ministry of the Environment,
Conservation and Parks of Ontario (http://www.airqualityontario.com/). For Mexico, $PM_{2.5}$
concentrations are derived from 27 stations of the Red Automática de Monitoreo Atmosférico (RAMA,
http://www.aire.cdmx.gob.mx/default.php?opc=%27aKBh%27) network.
All these stations are located within a circle of 50-km radius around the city centers of Paris, Toronto,
and Mexico City.

130       2.3. $NH_3$ and $PM_{2.5}$ from the GEOS-Chem model

We generate model outputs for March of 2011 because all three cities experienced both separate and
combined $PM_{2.5}$ and $NH_3$ pollution events during this period. We use version 12.7.2 of the GEOS-Chem
chemical transport model [Bey et al., 2001] driven by the MERRA-2 reanalysis product, including nested
domains over Europe and North America at a 0.5° × 0.625° horizontal resolution from which we extract
modeled surface values for each city. Boundary conditions for these two nested domains are created
using a global simulation for the same month at 2° × 2.5° resolution. Output for the analyzed month of
March includes monthly means, as well as hourly means for selected diagnostics, and is preceded by
two months of discarded model spinup time for the global simulation, and one month for each nested
run. Anthropogenic emissions are taken primarily from the global Community Emissions Data System
(CEDS) inventory [Hoesly et al., 2018], with regional emissions from the 2011 National Emissions
Inventory produced by the US EPA (NEI2011) used to override global values over the United States.
Biogenic non-agricultural ammonia, as well as ocean ammonia sources, are taken from the Global
Emission Inventories Activities database (GEIA, [Bouwman et al., 1997]). Open fire emissions are
generated using the GFED 4.1s inventory [Randerson et al., 2017]. Sulfate-nitrate-ammonium aerosol
processes are calculated using version 2.2 of the ISORROPIA thermodynamic module [Fountoukis and
Nenes, 2007]. Black carbon is handled as described in Wang et al. (2014), while secondary organic
aerosol is produced using the simplified irreversible scheme described in Pai et al., (2020).

148       2.4. Back-trajectories analysis from the HYSPLIT model

To determine the effect of long-range transport affecting the local air quality of the three cities, we
use the Hybrid Single-Particle Lagrangian Integrated Trajectory model (HYSPLIT, [Stein et al., 2015]).
Note that unlike the GEOS-Chem model, HYSPLIT does not include atmospheric chemistry. For the runs,
meteorological data are from the National Centers for Environmental Prediction (NCEP) / National
Center for Atmospheric Research (NCAR) reanalysis at 2.5-degree global latitude-longitude projection.
Note that visual inspection of the back trajectories shows that using a 2.5° resolution meteorological
dataset is similar to using a finer meteorological dataset at 1° resolution (GDAS). First, we run daily 24-
hour back-trajectories ending in the city-centers at the overpass time of the IASI instrument covering
the period 2008 to 2017. Then, for each day we calculate the mean of $NH_3$ total columns derived from
IASI observations in a 50-km radius circle around the cities associated with each back-trajectory.
Finally, all back-trajectories that are near to each other are merged in clusters and associated with the
corresponding local-scale IASI $NH_3$ concentrations.

161       2.5. ERA-5 meteorological data

The meteorological variables used in this study are extracted from the hourly ECMWF's reanalysis
(ERA5, [Hersbach et al., 2020]). ERA5 data are at 0.25° × 0.25° resolution (native horizontal resolution
of ERA5 is ~31km) and are interpolated in time and space to the IASI observation. The meteorological
parameters considered here are the skin temperature (Tskin, which is the physical temperature of the
Earth's surface), total precipitation (in meter of water equivalent - accumulated liquid and frozen
water, comprising rain and snow -) and relative humidity up to 2 meters above the surface calculated
from dew and air temperature at 2m from ERA5.
3. Results

170        3.1. $NH_3$ source regions identification and spatio-temporal variability over the Europe, North
America, and southern North America domains
Using 10-years of IASI observations, the main source regions of $NH_3$ in the 3 domains of study are
identified (Figure 2) and listed in Table 1. We identify 10, 9, and 19 $NH_3$ source regions over the Europe,
North America, and southern North America regions, respectively. All of the sources over the Europe,
North America domains are mostly related to agricultural practices (farming and spreading practices).
This is in agreement with previous calculation of worldwide nitrogen inputs from fertilizer and manure
[Potter et al., 2010]. Around southern North America, three sources are related to fertilizer or soda
ash industries (listed with C, G, O in Figure 2 and Table 1, [Van Damme et al., 2018]), the rest is
agricultural.
Spatio-temporal variabilities of $NH_3$ in the atmosphere in the three regions (Figure 2) are not expected
to be similar: $NH_3$ emissions from industries in the region of southern North America are released all
year long, whereas $NH_3$ emissions from agricultural practices (which are dominant over Europe and
North America), depend on various surface and meteorological conditions. In order to investigate this,
$NH_3$ concentrations using 10-years of IASI observations are assessed against atmospheric temperature
and precipitation derived from the ERA5 reanalysis over the three domains in Figure 3. It shows the
seasonal evolution of $NH_3$ from IASI over the three regions (left panel), along with the seasonal
evolution of temperature and precipitation (right panel).
For Europe and North America, $NH_3$ total columns are the highest in spring and summer. In fact, $NH_3$
concentrations over Europe exhibit two seasonal maxima in March/April and July/August
(supplementary material, Figure S1) and in North America the maxima are in May and September
(Figure S2). This is consistent with agricultural practices (i.e. fertilizer application) and higher air
temperature favoring $NH_3$ volatilization in the atmosphere.
The right panel of Figure 3 shows how temperature (red lines) and precipitation (blue bars) seasonally
evolve over the three regions. In winter, atmospheric temperatures are below 5 °C in Europe and North
America, and IASI observations reveal almost no $NH_3$ hot spots (left panel, Figure 3). This can be due
to the lack of $NH_3$ abundance, lower volatilization in this temperature range, no agricultural emissions
in winter and/or the reduced sensitivity of the IASI $NH_3$ retrievals in winter [Van Damme et al., 2017].
The high value over Canada and the Arctic in winter can be associated with high uncertainties in the
$NH_3$ retrievals due to low thermal contrast.
In southern North America, $NH_3$ seasonal variations are less pronounced than in the other two regions.
Figure 3 shows that the $NH_3$ concentrations over several sources, such as Torreon and San Juan de Los
Lagos (boxes B and F in Figure 2 right panel) are high during all seasons, which could be associated with
the weak seasonal cycle of temperature in this region closer to the equator.
In spring, seasonal precipitations are the lowest for the three regions. This is reflected in high $NH_3$
concentrations on the left panel. Over Europe and North America, this can be related to agricultural
spreading practices period and higher atmospheric temperature favoring $NH_3$ volatilization. In
southern North America, $NH_3$ concentrations observed by IASI are the highest in spring when
atmospheric temperatures are high and precipitations rates are low. In addition, biomass burning, that
are often encountered during this period could explain higher atmospheric $NH_3$ concentrations in
spring. $NH_3$ reach maximum values in April/May (Figure S3) just before the start of the rain season,
potentially reducing observed $NH_3$ concentrations due to the wet deposition of atmospheric gaseous
ammonia [Asman et al., 1998].
Since in Europe and North America $NH_3$ sources are mostly agriculture-related (with small
contributions from industries), the temperature/$NH_3$ relationship is expected to be relatively easy to
interpret: when the land surface temperature increases, volatilization of ammonia from the
fertilized/manured soil is favored, and atmospheric ammonia increases. The corresponding
determination factors $r^2$ for this relationship in Europe and North America are 0.85 and 0.80
respectively (polynomial fit of second order). This is not the case in southern North America, in which
some of the ammonia sources are also industrial and they contribute greatly to the atmospheric $NH_3$,
the concentrations of ammonia are therefore not directly temperature dependent, as we can see on
the right upper panel in Figure S4 ($r^2 = 0.46$). There is nonetheless a relationship in southern North
America that is due to the fact that we have constant high ammonia sources and temperatures (Figure
3). In fact, the relationships between $NH_3$ and temperature on one hand, and precipitation/relative
humidity on the other hand, are not linear; this has been equally shown in a previous study [Sutton et
al., 2013].
To further investigate the temperature/$NH_3$ relationship, we show in Figure 4 the evolution of $NH_3$
with respect to land surface temperature over different sub-regions of the Europe domain (listed in
Table 1). Similar Figures for the North America and southern North America domains are shown in the
supplement information (Figure S5 and S6). We observe a peak of $NH_3$ followed by a local maximum
plateau between 10 and 25°C approximately in all of the regions of the Europe domain (Figure 4). In
fact, the $NH_3$ detected in this range of temperature can indicate the fertilizer application period, since
most of them (up to 80%) were detected during the spring and fall seasons. For instance, over the Po
valley (region F in Table 1, Figure 4), 36% of the $NH_3$ detected in the bins 10 – 25°C correspond to the
spring season, whereas 35% correspond to the fall season (not shown here). In Celina-Coldwater
(region G in Table 1), 82% of the $NH_3$ detected between 10 and 25°C correspond to the spring and the
fall seasons, the percentage is split equally (Figure S5).
We choose to show the sub regions in the vicinity of the Europe domain, since they are mostly
agricultural sources. The "bumps" corresponding to the fertilizer application are very clear in all of the
sub-regions. This bump was detected to a lower extent for agricultural regions affecting North America
(supplementary material Figure S5). Over the agricultural regions in the southern North America
domain, the bumps are clear in the regions A to D (Figure S6, a). When the seasonal temperatures do
not fluctuate during the fertilizer application, any increase in atmospheric $NH_3$ is due to the sudden
addition of nitrogen fertilizers to the soil. In southern North America, the regions E to M show that the
highest $NH_3$ concentrations were observed as the temperature increased during the growing seasons
(Figure S6). A possible explanation to the resemblance among the regions A to D is that they share
similar climate properties (Steppe and Desert) unlike the rest of the sub-regions in the same domain
(tropical/subtropical). Since the temperatures in the Europe and southern North America domains are
higher (Figure 3, right panels) in spring and fall seasons (fertilizer application period) than those in
North America, this bump is clearer in the latter. The bumps seen for the Europe regional domain are
clearer than those of southern North America, possibly related to the fact that in autumn in Europe
precipitation is lower than those in southern North America, leading to lower $NH_3$ loss through wet
deposition.

## 3.2. NH₃ budget over the cities of Paris, Toronto, and Mexico

Temperature, relative humidity, and precipitation are not the only factors affecting the $NH_3$ concentrations. In order to analyze the impact of long-range transport on $NH_3$ concentrations measured over the cities (and not domains) of Paris, Toronto, and Mexico, HYSLPIT back-trajectories have been used. For each day of IASI $NH_3$ observations made in a 50-km radius circle from the city-center, a 24-hours back-trajectory has been performed from 2008 to 2017. There are between 3643 and 4008 back-trajectories for Paris, Toronto, and Mexico cities. Then, a seven-cluster analysis has been applied to these datasets and $NH_3$ mean concentrations measured inside the cities by IASI have been allocated to the different mean cluster trajectories according to the corresponding back-trajectories. Details about this analysis are described in the supplementary material Figure S7. The result is shown in Figure 5.

For Paris, 1/4 of all back-trajectories (875) that are associated with the highest $NH_3$ concentrations, i.e. $4.71x10^{15}$ molecules/$cm^2$ on average, are originating from the surrounding south regions (black line, Figure 5). Clusters 2 and 3 are also associated with high $NH_3$ concentrations and are coming from the north-northeast. This is consistent with previous analyses using HYSPLIT [Viatte et al., 2020] and FLEXPART models [Viatte et al., 2021]. As expected, the back-trajectories coming from the ocean are related with almost no $NH_3$ concentrations (light and dark blue lines, left panel).

Over Toronto, the highest $NH_3$ concentrations (24% and 14%) measured in the city are allocated to long-range transport located south-southwest (black and purple lines, middle panel) coming from the United-States where most of the feedlots are. 9 to 17% of $NH_3$ concentrations are coming from the west and the east of Toronto (cluster 3, 4, and 5) where atmospheric $NH_3$ have increased in the last decade possibly due to the reduced chemical loss of atmospheric $NH_3$ to form particulate $NH_4^+$ (Boxes A and B in Figure 2, [Yao and Zhang, 2019]). The 2 back-trajectory clusters that are related to low $NH_3$ concentrations are coming from the north (light and dark blue lines) where no $NH_3$ sources have been identified.

In the southern North America domain, back-trajectories are coming from relatively close regions since orographic conditions around Mexico-city limit long-range transport. In this city, the highest $NH_3$ concentrations are associated with air masses coming from the southwest (11%, black line, 22%, purple line, right panel) and southeast (27%, red line). Air parcels coming from the north are associated with relatively low $NH_3$ concentrations measured in Mexico City.

## 3.3. Pollution events over Paris, Toronto, and Mexico cities from 2008 to 2017

After assessing the $NH_3$ distribution under average climate conditions, we focus now on pollution events occurring at the three cities. These are identified by applying the Fourier series of order 3 [Yamanouchi et al., 2021; Herrera et al., 2022] on the surface $PM_{2.5}$ and satellite $NH_3$ observations at cities scale (i.e. 50-km radius circle from city-centers). The Fourier fit accounts for the "natural" variability (seasonality) in the time-series, and helps identify pollution events that are 2 standard deviations above this natural variability. It is a robust method commonly used to quantify trends and identify enhancements in long-term timeseries [Zellweger et al., 2009]. Pollution events occurrence per city per year (a) and per season (b) are shown in Figure 6.

The figure shows that $NH_3$ pollution episodes are found to be annually frequent at the three cities. In Toronto and Mexico cities, $PM_{2.5}$ pollution events are encountered annually (with higher number in Mexico) whereas no events are detected in 2009, 2015, and 2017 in Paris.

Numbers of identified days of PM$_{2.5}$ pollution events are 88, 58, and 50 in Mexico City, Toronto, and
Paris, respectively. For NH$_3$ pollution events, they occur more in Toronto than in Mexico City and Paris,
with number of days of 94, 73, and 56, respectively. Common days of high NH$_3$ and PM$_{2.5}$
concentrations are found in all three cities, especially in spring (Figure 6b), coinciding with the high
seasonal NH$_3$ concentrations shown in Figure 3.
To further investigate the impact of transport on pollution events occurring at the three cities, we have
analyzed the wind fields patterns for different cases: i) for the whole dataset (i.e. ensemble 2008-
2017), ii) for days of NH$_3$ and PM$_{2.5}$ pollution events occurred separately, and iii) for days when both
high concentrations are observed. Figure 7 shows wind roses computed for the ensemble and these
different types of pollution events (i.e. PM$_{2.5}$, NH$_3$ and both occurring during the same day). The radial
distance in the wind roses indicates the frequency of the wind direction occurrence. In general, wind
speed is lower at Mexico City (max 3 m s$^{-1}$) compared to Toronto and Paris (up to 10 m s$^{-1}$) because of
the mountainous topography that blocks and slows air masses exchange in Mexico.
In Paris, the ensemble wind-roses show a dominant northeast-southwest pattern. NH$_3$ pollution events
are associated with wind coming from various directions at all wind speeds which was suggested by
the HYSPLIT cluster analysis (Figure 5), whereas PM$_{2.5}$ events are present mainly under high northeast
wind. When both NH$_3$ and PM$_{2.5}$ high concentrations are observed in Paris, the wind field can have two
patterns: low wind speed coming from all direction (except from the south) or high wind speed coming
from the northeast. This confirms the importance of transport of NH$_3$ and PM$_{2.5}$ from the northeast
and could suggest the inter-conversion of PM$_{2.5}$ to NH$_3$ at low wind speed.
In Toronto, the ensemble show that dominant wind pattern is coming from the south. For all the
pollution events (NH$_3$, PM$_{2.5}$, and both) the wind is coming from the southwest, confirming the long-
range transport of pollutants from the United-States.
In Mexico City, the dominant pattern (ensemble) is southwest-northeast wind fields. For days of NH$_3$
pollution events, wind is mainly coming from the south-southwest, and for PM$_{2.5}$, wind come from all
direction with an important northeast wind pattern. Days of both pollution events are associated with
wind coming from the west-southwest only.

322         3.4. Case study: NH$_3$ and PM$_{2.5}$ concentrations comparison with the GEOS-Chem model in
March 2011
The occurrence of pollution events varies from year to year (Figure 6). However, in 2011, all three cities
experienced PM$_{2.5}$ and NH$_3$ separate and combined pollution events. For this reason, GEOS-Chem
model simulations were performed in March 2011 to interpret the events and evaluate the model
performance.
Spatial and temporal coincidence criteria have been applied to GEOS-Chem outputs to compare with
IASI morning observations, such as: model outputs between 8.30 and 11.30 AM coincident with IASI
overpasses have been selected, and only collocated model outputs (at 0.5° × 0.625° horizontal
resolution) have been selected coincident with IASI observations. Averages of numbers of coincident
observations are 1324, 1138, and 3000 over the Europe, North America, and southern North America
domains of study during March 2011.
Figure 8 shows the one-month comparison between the two datasets. Over the regional domains, the
coefficient of correlation between daily model NH$_3$ concentrations and IASI NH$_3$ observations are R =
0.50, R = 0.55, and R = 0.33, over Europe, North America, and southern North America, respectively,
with related p-values < 0.01. NH$_3$ columns derived from the GEOS-Chem model are overall

underestimated with Mean Relative Difference (MRD = (model - observations) / observations) of -37%, -31%, and -2% over Europe, North America, and southern North America, respectively.

Over Europe and North America, IASI and GEOS-Chem capture some of the same pollution events (on March 12, 15, and 30 over Europe, and March 12, 13, and 18 over North America). In southern North America, the underestimation of the GEOS-Chem $NH_3$ columns is less pronounced (MRD is -2%) than in the other regions but the day-to-day variability is not well represented in the model.

The GEOS-Chem model $NH_3$ total columns are lower than those from IASI in March 2011 over specific locations in the southern North America and Europe domains (Figure 8, right panels). For the Europe region, GEOS-Chem $NH_3$ columns are smaller than the IASI ones over the north of France, Belgium, the Netherlands, north of Spain (in particular sources A, B, C, D, E, I, and J of Figure 2) and the United Kingdom. For the southern North America domain, GEOS-Chem $NH_3$ columns are smaller than the IASI ones over the west Mexican coast (sources A, D, E, J of Figure 2/Table 1), central (source F, G, H of Figure 2) and southeast (sources O and P of Figure 2) of Mexico City and over the Pacific Ocean, whereas they are higher in Guatemala (source S, R of Figure 2), and West of Mexico City.

Over the North America domain, spatial distribution of the differences between $NH_3$ columns derived from GEOS-Chem and IASI are less pronounced than in the Europe and southern North America domains. IASI $NH_3$ columns are smaller than GEOS-Chem outputs over the south of the United-States and over the Lancaster County (sources E and I of Figure 2) and higher over Indiana in the United States.

At the city scales of Paris and Mexico City, the daily model $NH_3$ columns are in relatively good agreements with IASI observations within a 50-km radius circle from the city-centers (not shown here), since the coefficient of correlation are R = 0.42 and R = 0.52, respectively. Similar to the regional domains, GEOS-Chem $NH_3$ columns are relatively underestimated at the city scales of Paris and Mexico City, with a MRD of -108% and -28%. At the city scale of Toronto, the correlation between the $NH_3$ columns derived from the model and observed by IASI is poor, with a coefficient of correlation of R = -0.32, and a small underestimation of the modelled $NH_3$ concentrations is found with a MRD of -6%.

Local comparison of $PM_{2.5}$ concentrations at the city scale (over Paris, Toronto, and Mexico) is shown in Figure 9, left panels. They show that $PM_{2.5}$ concentrations calculated by the model in March 2011 are in relatively better agreement with the surface observations with R = 0.63, R = 0.43, and R = 0.54 in Paris, Toronto and Mexico City. In Paris and Mexico City, $PM_{2.5}$ concentrations values derived from the observations are overall higher than the GOES-Chem concentrations with MRD values of -13% and -20%, respectively, whereas GEOS-Chem $PM_{2.5}$ concentrations are higher than the observations in Toronto with MRD value of 519%.

The right panels of Figure 9 show the chemical composition of the $PM_{2.5}$ from GEOS-Chem. These inform us about the different pollution sources. Organic matter sources are splitted equally between the primary emissions and the oxidation of volatile organic compounds [Day et al., 2015]. SNA (sum of sulfate, nitrate, and ammonium) sources originate in chemical transformation of gaseous precursors in the atmosphere, whereas black carbon comes from primary emissions of industrial and traffic combustion.

According to the GEOS-Chem model, SNA dominates the $PM_{2.5}$ chemical composition mass in March 2011 over the three cities, meaning that the dominant source of $PM_{2.5}$ mass comes from the secondary oxidation path. This partition of SNA in March 2011 from the model is higher than what have been reported based on 1-year-measurements performed in 2013: 43%, 42%, and 33% of the $PM_{2.5}$ mass composition in Paris, Toronto, and Mexico City, respectively [Cheng et al., 2016].

In Paris, the March 2011 pollution episode has been analyzed in terms of geographical origins and
aerosol properties [Chazette et al., 2017] but not in terms of aerosol speciation.
Comparing the GEOS-Chem outputs used in this study with two years of observations of aerosol
chemical composition in Paris (2011-2013) [Petit et al., 2015], we found that the sulfate component is
slightly higher in the GEOS-Chem model than in the springtime observations (21% compared to 11%)
whereas modelled organic carbon is lower than the observations (8% compared to 33%). This
springtime underestimation of organics in atmospheric models has previously been reported in Paris
[Sciare et al., 2010; Petit et al., 2015; Lanzafame et al., 2021] and could be associated with an
underestimation of the organic matter emissions from residential contributions [Van der Gon et al.,
2015].  Regarding the secondary aerosol, observations in Paris during the March 2015 pollution event
show that it accounts for more than 50% of the PM concentration [Petit et al., 2017], which is in
agreement with the SNA partition in our GEOS-Chem model simulation.
In Toronto, $PM_{2.5}$ speciation is monitored by the National Air Pollution Surveillance Program (NAPS,
https://www.canada.ca/en/environment-climate-change/services/air-pollution/monitoring-
networks-data/national-air-pollution-program.html) network. Observations in March 2011 reveal that
inorganic nitrate burden is overestimated by a factor 2 in the GEOS-Chem run (41% in the model
compared to 20% in the observations), whereas sulfate and black carbon abundances are
underestimated by a factor 2 (15 and 6% in the model compared to 27 and 12% in the observations).
In Mexico City, the organic matter represents the most abundant fraction of the aerosol, which is
consistent with measurements made during several campaigns performed in the dry season of 2006
during the Megacity Initiative: Local And Global Research Observations (MILAGRO, [Molina et al.,
2010]) and Aerosoles en Ciudad Universitaria (ACU) in 2015 [Salcedo et al., 2018]. Observations
performed during the dry-warm season of 2019 reported that SNA correspond to 30% of the aerosol
mass concentration [Retama et al., 2022], which is consistent with what has been reported before
[Cheng et al., 2016] and the chemical composition modelled in our study. The organic fraction is found
to be dominant in the observations [Retama et al., 2022] as suggested in the GEOS-Chem model over
Mexico-City. Daily cycles appear overexaggerated in the model with maxima well represented and
minima greatly underestimated. This could suggest model issues in term of atmospheric dynamics
(removal/transport or planetary boundary layer dynamics) due to coarseness of grid.
4. Conclusion
The AmmonAQ project aims to determine the impact of intensive agricultural practices on urban
pollution in the Paris, Toronto, and Mexico metropolitan areas. For this purpose, $PM_{2.5}$ and $NH_3$
measurements from in situ instruments and satellite infrared spectrometers, and atmospheric model
simulations, have been combined.
Using 10-years of IASI observations, $NH_3$ sources regions have been identified. All of the sources are
from the agricultural sector (husbandry and fertilizer application) in the Europe and North America
domains, whereas, some of them are industrial in the southern North America region. Consequently,
the spatio-temporal variability of $NH_3$ is different, with stronger seasonal variabilities in Europe and
North America. A strong correlation is found between $NH_3$ total columns and surface temperature
(Tskin) for all regions, with higher correlation over agricultural regions, and when the temperature
seasonal cycle is pronounced. We find that the timing of the fertilizer application can be detected
through local maxima in the $NH_3$/Tskin relationship curve.
According to HYSPLIT cluster analysis, the highest $NH_3$ concentrations measured at the city scales are
associated with air masses coming from the surrounding regions: the north-northeast of Paris, the

south-southwest of Toronto, and the southeast/southwest of Mexico City. These lead to the exacerbation of the degradation of air quality in each of the three cities.

Pollution episodes are found to be annually frequent at the three cities, especially in springtime when high $NH_3$ and $PM_{2.5}$ are observed. In Paris and Mexico pollution is transported along the northeast-southwest line, whereas, in Toronto, the transboundary transport of pollutant from the United-States is dominant during pollution events.

The evaluation of the GEOS-Chem outputs in March 2011 reveals that $NH_3$ concentrations are overall underestimated by the model at the regional scale, with, however, a good representability of the day-to-day variability in Europe and North America domains. $NH_3$ columns derived from IASI and the GEOS-Chem model exhibit substantial spatial differences in the Europe and southern North America areas. In term of $PM_{2.5}$ concentrations at the city scales, we show that they are underestimated in Paris and Mexico, but overestimated in Toronto.

The IASI thermal infrared remote sensors have proved to be valuable to monitor pollution events over cities. The main limitations are associated with the low revisit time (at the beginning and at the end of the day), the lack of sensitivity to the surface in particular in winter, and some areas are not well covered during cloudy scenes. In the near future the next generation of instruments will have improved capabilities to sound deeper in the atmosphere [Crevoisier et al., 2014]. The IRS-MTG satellite instrument that should be launched in 2024 in geostationary orbit will offer the capacity to enhance this research over Europe thanks to better temporal (measurements every 30-45 minutes) and spatial (4 km x 4 km pixel) resolution.

Data availability

The near-real-time IASI NH3 (ANNI NH3-v3) data used in this study are freely available through the Aeris database https://iasi.aeris-data.fr/nh3-i/ (Van Damme et al., 2021) (last access: 1 April 2022). All hourly observations of $PM_{2.5}$ concentrations are available from the Airparif network (https://data-airparif-asso.opendata.arcgis.com/), the Ministry of the Environment, Conservation and Parks of Ontario (http://www.airqualityontario.com/), and the Red Automática de Monitoreo Atmosférico (RAMA, http://www.aire.cdmx.gob.mx/default.php?opc=%27aKBh%27) network (last access: 1 April 2022). The GEOS-Chem outputs are currently available upon request. All MATLAB/PYTHON codes used to create any of the figures and/or to create the underlying data are available on request.

Author contributions

CV, CC, SY, and KS designed the AmmonAQ project. MV and LC provided the IASI data. WP provided the GEOS-Chem outputs. CV and RA analyzed the data. CV, RA, and SS wrote the manuscript draft. BH, MG, KS, P-FC, and CC reviewed and edited the manuscript.

Competing interests

The authors declare that they have no conflict of interest.

Acknowledgments

AmmonAQ results from a joint research program between CNRS (National Center for Scientific Research of France) and the University of Toronto which funded one year of common research in 2019. Research at ULB was supported by the Belgian State Federal Office for Scientific, Technical and Cultural Affairs (Prodex HIRS) and the Air Liquide Foundation (TAPIR project). LC is Research Associate

supported by the Belgian F.R.S.-FNRS. This project has received funding from the European Research
Council (ERC) under the European Union's Horizon 2020 and innovation programme (grant agreement
No 742909, IASI-FT advanced ERC grant). The MERRA-2 data used in this study have been provided by
the Global Modeling and Assimilation Office (GMAO) at NASA Goddard Space Flight Center.

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

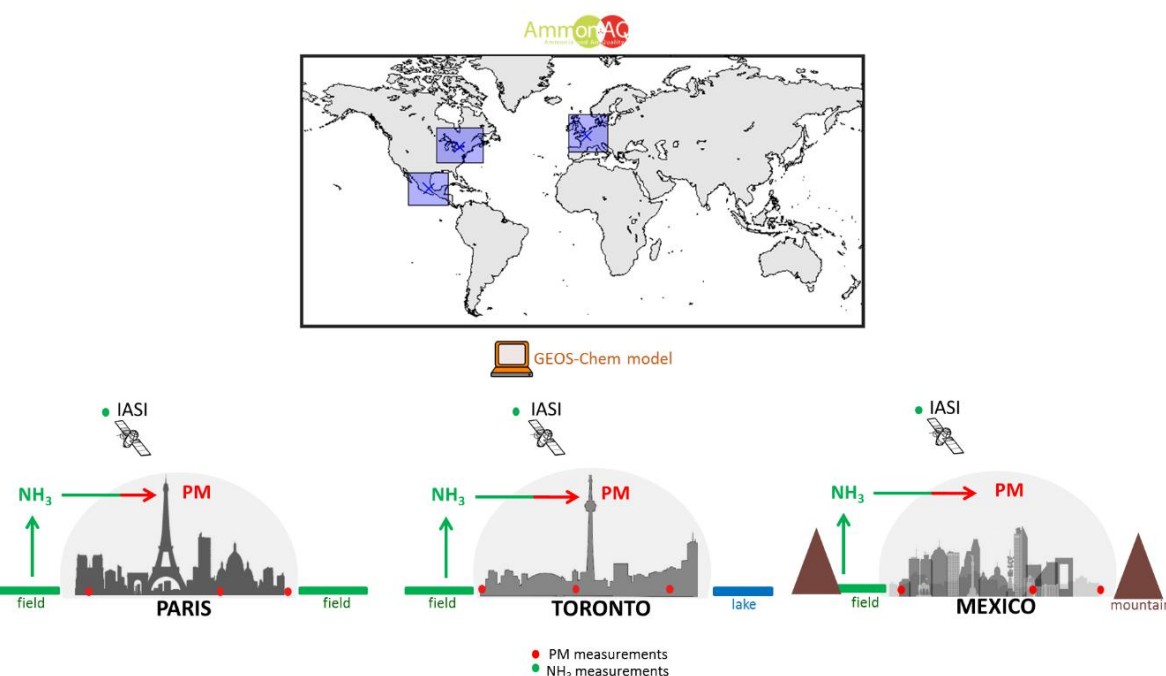


Figure 1: schematic representation of the AmmonAQ project. Upper panel: the three study regions investigated (in blue rectangles). Lower panel: presentation of each city and regional domain and different datasets used.


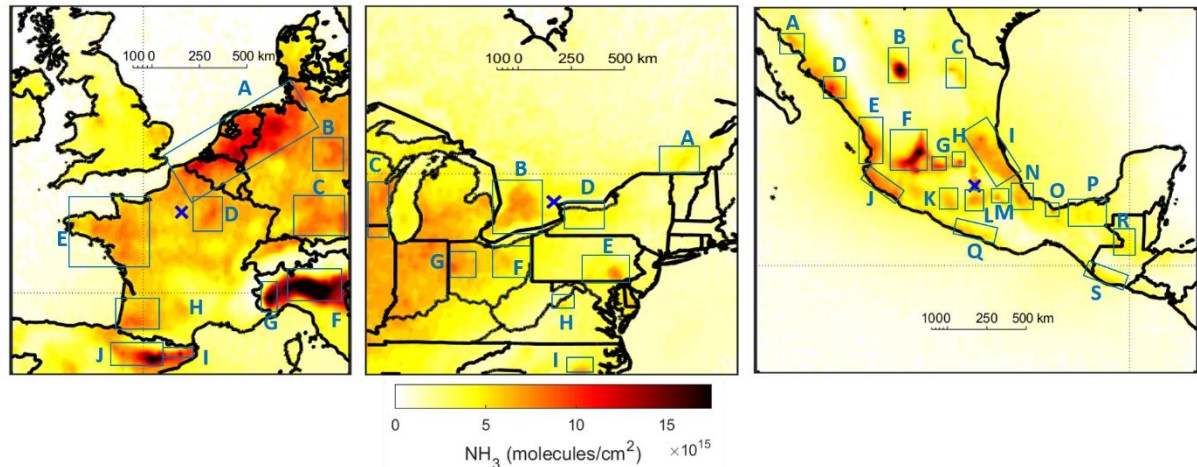

Figure 2: Source region identification of NH₃ derived from 10 years average of IASI total columns (molecules/cm²)
from 2008 to 2017. The blue crosses indicate Paris, Toronto, and Mexico cities locations.

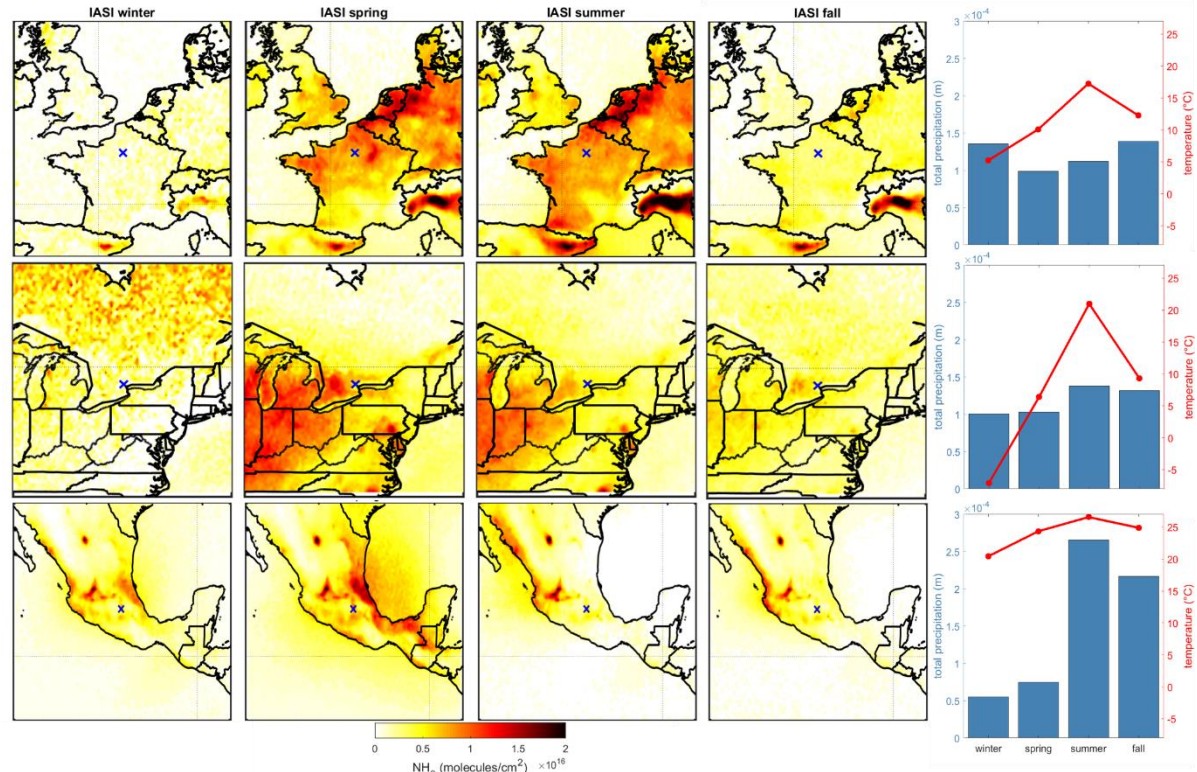


Figure 3: Seasonal maps of NH₃ total columns (molecules/cm²) derived from 10 years (2008-2017) of IASI observations, along with seasonal means of atmospheric temperature (red line) and precipitation (blue bar chart) over the Europe (upper panels), North America (middle panels), and southern North America (lower panels) regions.


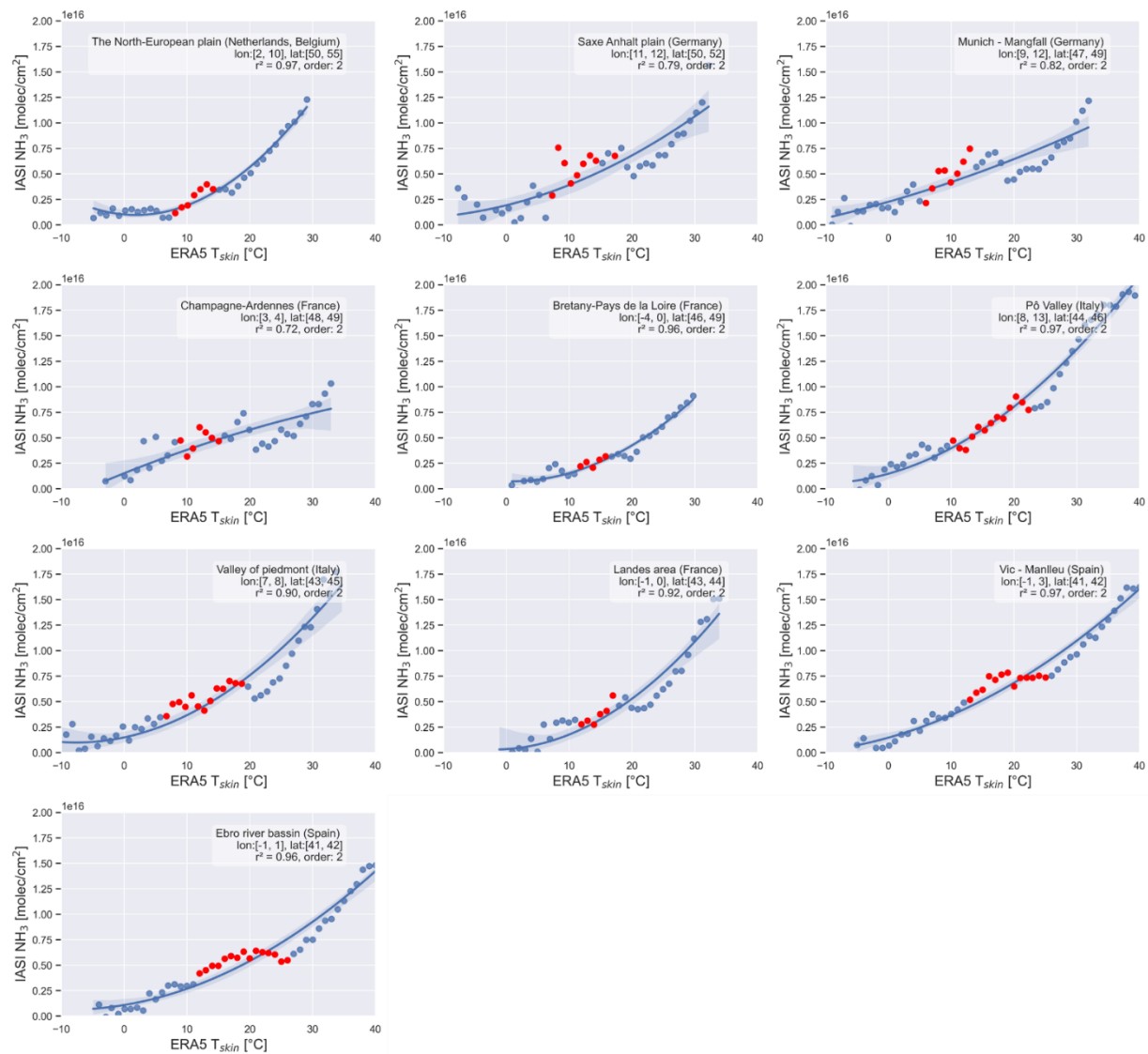


Figure 4: Yearly IASI NH₃ total columns (molecules/cm²) averaged per bins of ERA5 skin temperatures (°C), with
an interval of 1°C between each consecutive bin. The red circles denote the growing seasons, at least 60% of the
NH₃ is detected during March-May and Sept-Nov periods. See Figure 2 and Table 1 for the localization of the sub-
regions around Europe.

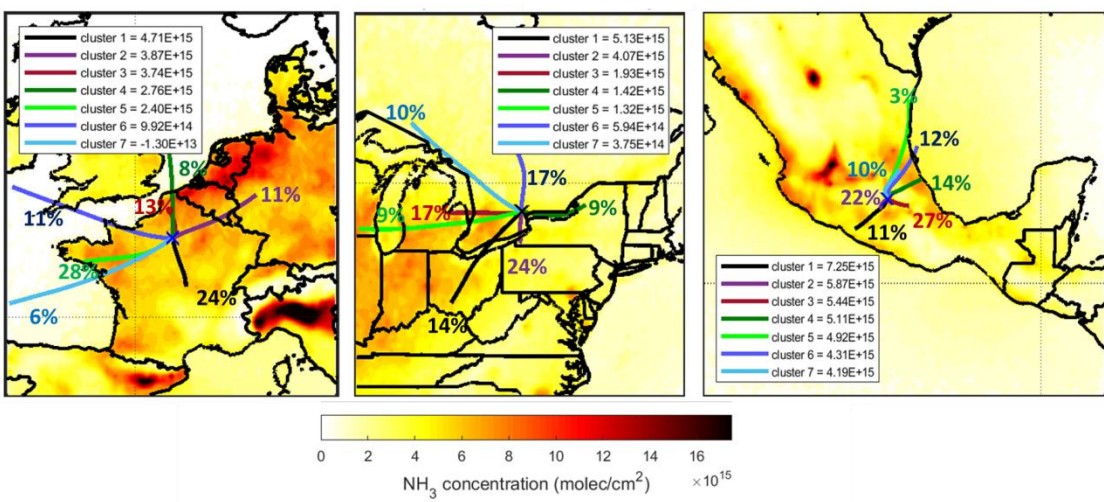

Figure 5: Seven cluster–mean backward trajectories over the Europe, North America, and southern North America regions for the whole time period between 2008 and 2017. Back-trajectories are color-coded in function of the corresponding NH₃ concentrations measured inside the cities. The numbers indicate the percentage of trajectories allocated to a cluster.

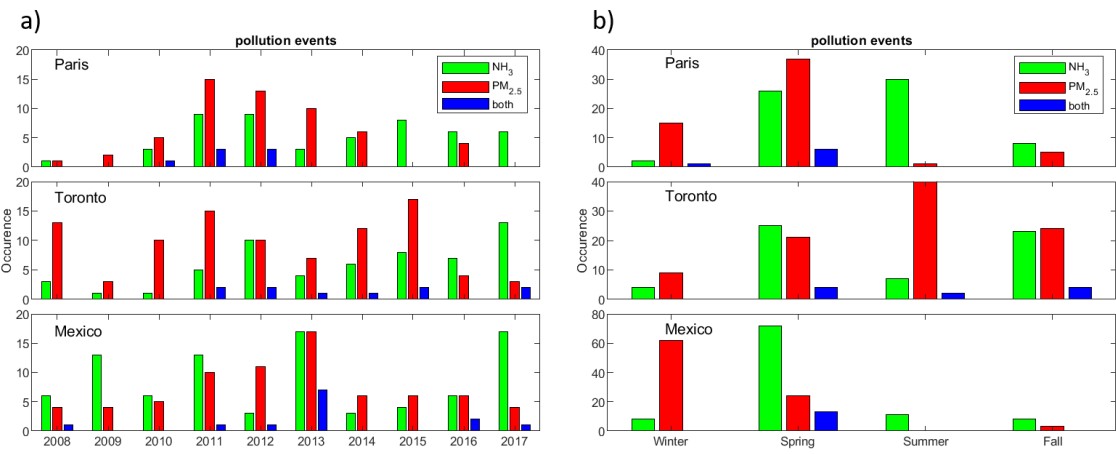


Figure 6: Annual (a) and seasonal (b) occurrence of pollution events of $NH_3$ (green bars), $PM_{2.5}$ (red bars), and
$NH_3$ and $PM_{2.5}$ simultaneous (blue bars) detected from 2008 to 2017 in Paris (upper panel), Toronto (middle
panel), and Mexico (lower panel) cities.

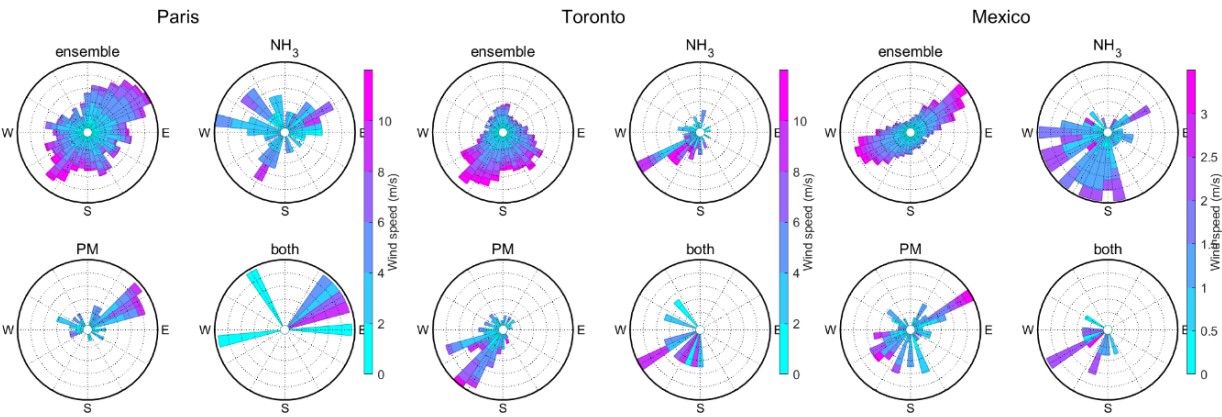


Figure 7: Wind roses corresponding to the ensemble of all observations, the $NH_3$, $PM_{2.5}$, and both $NH_3$ and $PM_{2.5}$
simultaneous pollution events derived from 2008 to 2017 over Paris (left panels), Toronto (middle panels), and
Mexico (right panels) cities.

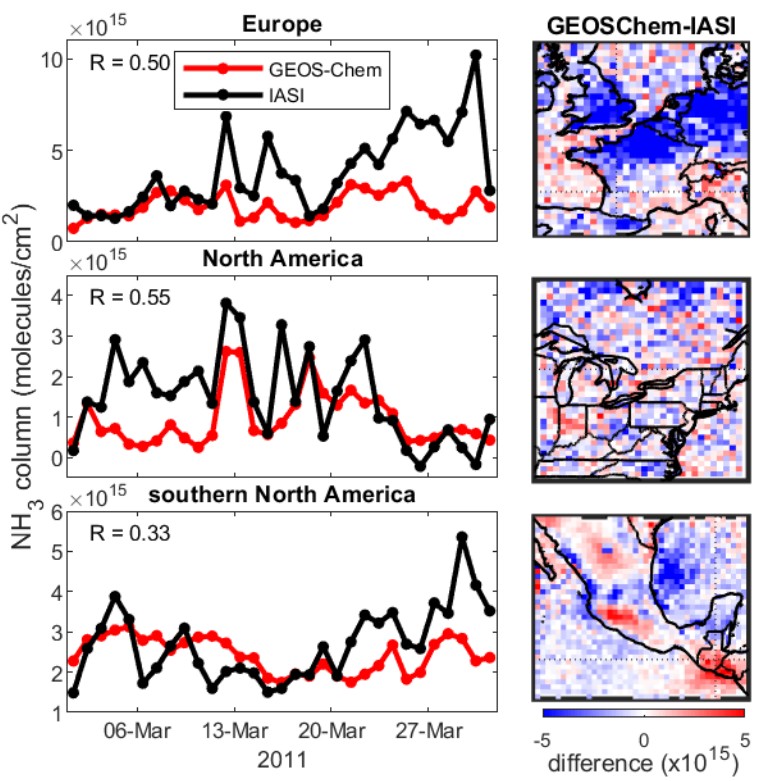


Figure 8: Left: time-series of daily NH$_3$ columns derived from IASI (black lines) and the GEOS-Chem model (red lines) over Europe (upper panel), North America (middle panel), and southern North America (lower panel). Right: maps of NH$_3$ columns (in molecules/cm$^2$) differences between IASI and GEOS-Chem model (model-observations) for March 2011.


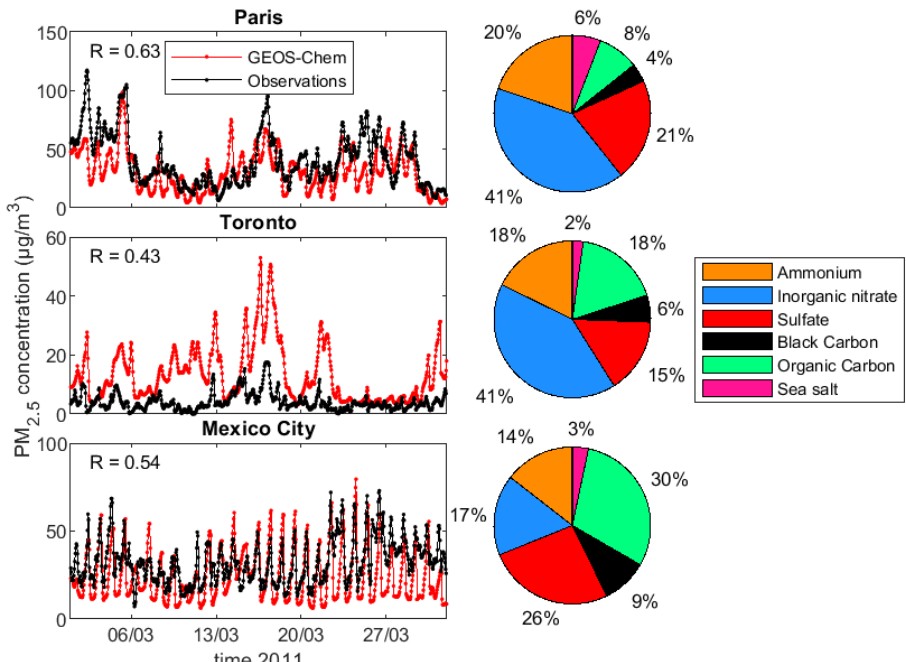


Figure 9: left: Time-series of hourly PM$_{2.5}$ (µg/m$^3$) derived from surface observations (black lines) and the GEOS-Chem model (red lines) over Paris (upper panel), Toronto (middle panel), and Mexico (lower panel) cities for March 2011. Right: PM$_{2.5}$ speciation (% in total mass) derived from the GEOS-Chem run for March 2011.

Table 1: List of NH$_3$ source regions identified using 10-years average of IASI total columns (molecules/cm²) over
the Europe, North America, and southern North America regions.

| | Europe [41°-59°N ; -11.25°- 16.25°E] | | North America [35°-53°N ; 93.75°-63.75°W] | | Southern North America [9°-29°N ; 113.75°-86.25°W] |
|---|---|---|---|---|---|
| A | The North-European plain[1,2] | A | Granby (Canada) | A | Obregon (Mexico)[1] |
| B | Saxe Anhalt plain (Germany) | B | Elmira-Kitchener-Guelph (Canada) | B | Torreon (Mexico)[1,2] |
| C | Munich - Mangfall (Germany) | C | Brillion area (U.S.A.) | C | Garcia (Mexico)**[1] |
| D | Champagne-Ardennes (France) | D | New-York state (U.S.A.) | D | Culiacancito (Mexico)[1,2] |
| E | Bretany-Pays de la Loire (France)[2] | E | Lancaster county (U.S.A.) | E | Nayarit (Mexico) |
| F | Pô Valley (Italy)[1,2] | F | Wayne county (U.S.A.) | F | Jalostotitlan-San Juan de Los Lagos (Mexico)[1,2] |
| G | Valley of piedmont (Italy)[1,2] | G | Celina-Coldwater (U.S.A.)[1] | G | Salamanca – Villagran (Mexico)*[1] |
| H | Landes area (France) | H | Shenandoah Valley-Bridgewater (U.S.A.)[1] | H | Ezequiel Montes (Mexico)[1,2] |
| I | Vic - Manlleu (Spain)[1,2] | I | Lenoir County (U.S.A.) | I | Tampaon, Loma Alta (Mexico)[1] |
| J | Ebro river bassin (Spain)[1,2] | | | J | Tecoman (Mexico) |
| | | | | K | Coyuca de Catalan (Mexico) |
| | | | | L | Morelos (Mexico) |
| | | | | M | Tochtepec-Tehuacan (Mexico)[1] |
| | | | | N | South of Veracruz (Mexico) |
| | | | | O | Cosolaecaque (Mexico)*[1] |
| | | | | P | Tabasco (Mexico) |
| | | | | Q | Guerrero (Mexico) |
| | | | | R | Chisec (Guatemala) |
| | | | | S | Texcuaco (Guatemala) |

*Fertilizer industry ** Soda ash industry
[1] Van Damme et al., 2018; Clarisse et al., 2019
[2] Dammers et al., 2019
