# Peer review of "Camille Viatte1, Rimal Abeed1, Shoma Yamanouchi2,3, William Porter4, Sarah Safieddine1, Martin Van"

_EGUsphere, 2022_

## Referee Comment (RC2)

**Review of "NH$_3$ Spatio-temporal variability over Paris, Mexico and Toronto and its link to PM$_{2.5}$ during pollution events"**

**Summary**

This paper analyses 10 years of IASI NH3 data over three large domains, each of which encompasses a major metropolitan area (Paris, Toronto and Mexico City). The ten year average and seasonal means of the NH$_3$ total columns from the IASI instruments deployed on Metop-A and Metop-B are presented and the relationships between NH3 amounts and temperature and precipitation are evaluated; the authors find a strong correlation between temperature and NH3 amounts in the Paris and Toronto domains, but only a weak one in the Mexico City domain. This analysis is extended to sub-domain scales, focusing on a number of previously identified source regions, and expanded to include the effect of relative humidity. Many of the smaller domains show an interesting local maxima in the NH$_3$ vs temperature plot that coincides with fertilization activities.

The impact of wind direction on air quality is also examined, using back-trajectories from HYSPLIT and a cluster analysis, along with in situ PM$_{2.5}$ data from local networks. The PM$_{2.5}$ data is also used in conjunction with the NH$_3$ columns to identify and count pollution events; wind roses are then constructed to determine the wind patterns on days with high pollution; interestingly PM$_{2.5}$ and NH$_3$ are not always high on the same days. Finally IASI NH$_3$ and the local PM$_{2.5}$ data are compared against GEOS-Chem output for one month. This last analysis is interesting but too limited to provide really useful information on GEOS-Chem performance.

The paper is well laid out and clearly written. The plots are of high quality and in general easily understood, though a few require more detailed captions, described below. It requires only some minor edits and clarifications to be acceptable for publication. Overall, a good illustration of how to apply a number of different techniques and data sources to the problem of understanding the drivers of pollution events.

**Technical comments**

Section 3.1: How are the IASI averaged and what is the grid resolution? How are the boxes doe each source region defined?

Figure 3: Please comment on the high NH$_3$ values over the Arctic.

Lines 266-274: Is each back-trajectory associated with the 50 km NH$_3$ mean for that day? Could the authors please briefly describe the clustering approach? Are the NH$_3$ means clustered according to the corresponding back-trajectories?

Line 289: Which sources have increased near Toronto?

Line 303: What are the criteria used for defining a pollution event?

Line 313: It would be useful to add the spring plot here

Line 319: What does the radial distance in the wind roses in Figure 7 indicate?

Line 350: Please state the coincidence criteria.

Lines 360-364: I don't agree that the GEOS-Chem and IASI are in good agreement; the authors should state that the two datasets capture some of the same pollution events.

Figure S4: Are the bins for the RH plot also specified by temperature? This does not seem right.

**Minor edits**

Lines 32-33: … in Paris and Mexico **City;**

Line 65: However, NH3 concentrations are increasing in many countries: France, Canada and Mexico reported increases of ……

Line 67: … are composed **of** …

Line 72: …nitrate formed **from** …

Line 127: … are located **within** a 50-km

Line 134: … the same **month**

Line 150: **Note** that

Line 153: … daily 24-**hour** back-trajectories …

Line 156: Finally, all back-trajectories are combined …. (not sure here what the authors mean to say)

Line 182: … practices (**which are dominant** over Europe and North America)

Lines205: … **in this region closer to the Equator**

Line 220: …(with **small contributions** from industries)

Line 255: …nitrogen fertilizers **to**

Line 267: concentrations. **In order to** analyze …
Line 280: … associated with the **highest** …

Line 281: …on average, are **originating** …

Line 322: … concentrations are **observed**

Line 352: …numbers of **coincident** observations

Line 403:  …sources **are**

Line 440: … surrounding regions**:**

**Line 441: …These lead**

Line 444: … and Mexico **pollution is transported along the northeast-southwest line**,

Line 458: … **launched** in

---

## Author Response (AR1)

**Review 1 of "NH3 Spatio-temporal variability over Paris, Mexico and Toronto and its link to PM2.5 during pollution events"**

**Referee: General comments:**

The manuscript of egusphere-2022-413 presented the result of a unique project of the AmmonAQ that targeted different three areas of Paris, Toronto, and Mexico City. Towards better air quality by mitigating NH3 emissions, the finding of this study will contribute to improving the atmospheric environment. I generally agree with this manuscript being published; however, some points are concerned. Especially, I am wondering about the uncertainty of the satellite NH3 measurement dataset analyzed in this study. Please address the following specific points.

Authors: We would like to thank the referee for this positive review and all the relevant comments that we have addressed in the following document.

**Specific comments:**

P3, L52: Please specify these five countries.

The five countries are United-States, China, Netherlands, United-Kingdom, and Canada. We have included this list in the revised manuscript.

P3, L65: Are these increasing trends explained by the expansion of NH3 emissions, or meteorological variability (e.g., temperature)?

In Europe, $NH_3$ emissions are relatively stable (with a decrease of 4 % between 2008 and 2012 followed by an increase of 3% between 2013 and 2018). However, IASI observations reveal an increase that could be associated with meteorological conditions, with high temperature and drought recorded in 2018.

In Canada, emissions are also relatively constant between 2008 and 2018, but biomass burning sustains increased $NH_3$ emissions in the northern hemisphere.

In Mexico, the EDGAR inventory shows an increase in $NH_3$ emissions.

These conclusions must be tempered by the fact that the lifetime of $NH_3$ in the atmosphere increases with decreasing concentrations of nitrogen oxides and sulfur.

We have added a sentence in the revised manuscript to explain the trends: "These trends are likely explained by increasing emissions, partly due to increased temperature (Europe) and biomass burning (Canada). However, decreasing concentrations of nitrogen and sulfur oxides e.g. in Europe and China also increase the ammonia atmospheric lifetime and plays a role in the reported upward trends."

P3, L77: Does "the standard" indicate the standard in Mexico? Because this study conducted the comparison over three regions, it will be better to explicitly state it.

We have included 'national' standard to explicitly state it. In addition, we have modified the standard value for Mexico because it has recently changed in October 2021. Therefore, we have modified the sentence as follows: "In Mexico, $PM_{2.5}$ concentrations often exceed the national standard of 41 µg/m$^3$ for the 24-hour mean [NOM-025-SSA1-2021, 2021]".

P5, L111 (Section 2.1): Because of the recent progress in satellite $NH_3$ measurement, I would like to strongly suggest including the discussion of the uncertainty of satellite data, such as the detection limit (https://doi.org/10.5194/acp-19-12261-2019).

In this section of the revised manuscript, we have added description about the detection limit: "The detection limit depends both on the atmospheric state (mainly thermal contrast and $NH_3$ abundance) and the instrument characteristics. For IASI, the minimum detection limit is found to be 4-6x10$^{15}$ molecules/cm$^2$ [Clarisse et al., 2010]."

What is the mottled pattern found over Canada during winter in Figure 3?

The sensitivity of IASI measurements is intimately related to the thermal contrast between the surface and the first layers of the atmosphere [Clerbaux et al., 2009]. When the detection is possible (with good thermal contrast), the peak sensitivity for $NH_3$ is in the boundary layer [Clarisse et al., 2010]. The high value over Canada and the Arctic in winter can be associated with high uncertainties in the $NH_3$ retrievals due to low thermal contrast and high emissivity from snow. We have added this sentence in the revised manuscript to clarify: "The high value over Canada and the Arctic in winter can be associated with high uncertainties in the $NH_3$ retrievals due to low thermal contrast".

Can all satellite measured NH3 close to zero be used in Figure 4?

Yes, removing or keeping these values won't change the relationship between temperature and the ammonia averages, therefore the discussion of the following figure will not be affected. In Figure R1, we show the same plots as Figure 4 (in the manuscript) but with the values of $NH_3 \leq 0.25 \times 10^{16}$ molecules cm$^{-2}$ removed. Note that the r² did not change much, in the presence and absence of these values.

[Figure]

Figure R1. Same as Figure 4 in the revised manuscript but with NH$_3$ average $\leq 0.25 \times 10^{16}$ molecules cm$^{-2}$ removed.

Is it available AK when comparing GEOS-Chem? The information on AK and how to calculate it in the comparison with the model is not described.

The IASI retrieval algorithm for NH$_3$ does not provide averaging kernels. We have qualitatively compared IASI NH$_3$ columns with GEOS-Chem NH$_3$ columns using all the morning IASI measurements available, following the recommendation provided in Van Damme et al. [2017]. Previous studies have demonstrated good agreement with surface and FTIR measurements (e.g., Clarisse et al., 2010; Van Damme et al., 2015; Dammers et al., 2017; Viatte et al., 2021; Van Damme et al., 2021), demonstrating that IASI retrievals are sensitive to NH$_3$ located in the lower layers of the atmosphere.

P5, L115: What is the actual gridded data (e.g., Figure 2) analyzed in this study?

We have gridded the IASI data at 0.25° x 0.25° degrees. This information has been included in the revised manuscript as follow: "In this work, we use version 3 of the ANNI-NH$_3$ product [Van Damme et al., 2021; Guo et al., 2021] from IASI Metop-A/B morning overpasses over the period 2008 to 2017 gridded a spatial resolution of 0.25° x 0.25°".

P5, L130: Although we can find the reason to choose the model simulation period of 2011 in P12, L344-P13, L349, it is better to be shortly explained here.

We have added a short sentence in this section: "because all three cities experienced both separate and combined PM$_{2.5}$ and NH$_3$ pollution events during this period."

P6, L178: Are these three panels shown with the same horizontal distance? If different, the scaler might be helpful.

The three panels are approximatively shown at the same horizontal distance, but we have inserted a scaler in each panel to help the reader.

[Figure]

Figure R2: Same as Figure 2 in the revised manuscript.

P6, L152: Too coarse reanalysis resolution to investigate air mass trajectories on 50 km radius-circle at each city?

We have run 24-hour HYSPLIT back-trajectories every day for the whole IASI dataset from 2008 to 2017. The idea is to determine the effect of long-range transport affecting air quality within the cities during this decade. Running 3652 daily back-trajectories (10 years) using a finer meteorological reanalysis would have been very time consuming. In addition, results derived from a finer resolution meteorological dataset are very similar. We have performed a test for a sub dataset (July 2008) running 24h-backtrajectories ending in Paris using NCEP at 2.5° resolution and GDAS at 1° resolution (Figure R1). Results show no significant differences when using a finer meteorological dataset. We have inserted in the revised manuscript a sentence to address this concern: "Note that results using a finer meteorological dataset (GDAS at 1° resolution) show no significant differences."

[Figure]

*Figure R3: 24-h HYSPLIT back-trajectories ending in Paris using the meteorological dataset of GDAS (at 1° resolution, left panel) and NCEP (at 2.5° resolution, right panel) for July 2008.*

P7, L188: Same to Europe and southern North America, source information of "(Canada)" or "(U.S.A.)" can be useful in this Table 1.

We have added information about the sources countries in the revised Table 1 as follows:

| Europe [41°-59°N ; -11.25°- 16.25°E] | | North America [35°-53°N ; 93.75°-63.75°W] | | Southern North America [9°-29°N ; 113.75°-86.25°W] | |
|---|---|---|---|---|---|
| A | The North-European plain[1,2] | A | Granby (Canada) | A | Obregon (Mexico)[1] |
| B | Saxe Anhalt plain (Germany) | B | Elmira-Kitchener-Guelph (Canada) | B | Torreon (Mexico)[1,2] |
| C | Munich - Mangfall (Germany) | C | Brillion area (U.S.A) | C | Garcia (Mexico)**[1] |
| D | Champagne-Ardennes (France) | D | New-York state (U.S.A) | D | Culiacancito (Mexico)[1,2] |
| E | Bretany-Pays de la Loire (France)[2] | E | Lancaster county (U.S.A) | E | Nayarit (Mexico) |
| F | Pô Valley (Italy)[1,2] | F | Wayne county (U.S.A) | F | Jalostotitlan-San Juan de Los Lagos (Mexico)[1,2] |
| G | Valley of piedmont (Italy)[1,2] | G | Celina-Coldwater (U.S.A)[1] | G | Salamanca – Villagran (Mexico)*[1] |
| H | Landes area (France) | H | Shenandoah Valley-Bridgewater (U.S.A)[1] | H | Ezequiel Montes (Mexico)[1,2] |
| I | Vic - Manlleu (Spain)[1,2] | I | Lenoir County (U.S.A) | I | Tampaon, Loma Alta (Mexico)[1] |
| J | Ebro river bassin (Spain)[1,2] | | | J | Tecoman (Mexico) |
| | | | | K | Coyuca de Catalan (Mexico) |
| | | | | L | Morelos (Mexico) |
| | | | | M | Tochtepec-Tehuacan (Mexico)[1] |
| | | | | N | South of Veracruz (Mexico) |
| | | | | O | Cosolaecaque (Mexico)*[1] |
| | | | | P | Tabasco (Mexico) |
| | | | | Q | Guerrero (Mexico) |
| | | | | R | Chisec (Guatemala) |
| | | | | S | Texcuaco (Guatemala) |

*Fertilizer industry ** Soda ash industry
[1] Van Damme et al., 2018; Clarisse et al., 2019
[2] Dammers et al., 2019

P13, L358: Should the denominator be "observations" when comparing observation and model? Why model is referred to as a criterion?

We agree with the referee that the model is not referred here as a criterion. Therefore, we have changed the denominator to be observations. The result doesn't change much the MRD, so the discussion remains valid.

P13, L359: Are these values positive? If model underestimation, are these negative?

To calculate the Mean Relative Difference (MRD), we have revised the formula to be more intuitive as follows: MRD = (model - observations) / observations) as suggested above. Thus, when the MRD is positive, model data are higher than the observations. Conversely, when the MRD is negative, model data are smaller than the observations. In the latter case, a negative MRD means that the model underestimate compared to the observations.

P14, L379: From the spatial mapping over Europe, this seems to be simply led to model overestimation, and this is not consistent with the timeseries and relevant discussion in the main text. Please confirm this figure.

In the revised figure (see figure R4), we choose to be consistent and show (GEOSChem-IASI) NH₃ columns. Therefore, if this difference is negative (blue colour in the figure), then IASI NH₃ columns are higher than the model ones, leading to an underestimation of the model. To clarify, we have added "(model- observations)" in the caption of the revised manuscript.

[Figure]

*Figure R4: same as Figure 8 in which we choose to show the difference (model-observations) to be consistent throughout of the revised manuscript.*

P15, L413-418: From this comparison on PM2.5 component, I am suspicious about the result in other cities of Paris and Mexico City. When we considered this poor performance for PM2.5 components, the result in Figure 9 and the relevant discussion seems to be meaningless. Is this performance for PM2.5 component useful (worse or better than other studies)? If not, I would like to request to reconsider this final section in P14, L391-P16, L425.

The component comparison is important in Toronto because it helps explain why the model is performing so poorly there.

Unfortunately, to our knowledge, there are no PM$_{2.5}$ chemical composition observations available to compare individual components in Paris and Mexico City in March 2011. However, we found in the literature relevant results that could strengthen the final discussion.

In Paris, the March 2011 pollution episode has indeed been analyzed in terms of geographical origins and aerosols properties [Chazette et al., 2017] but not in term of aerosol speciation.

[revised manuscript text omitted]

**Referee :**

**Summary**

This paper analyses 10 years of IASI NH3 data over three large domains, each of which encompasses a major metropolitan area (Paris, Toronto and Mexico City). The ten year average and seasonal means of the NH3 total columns from the IASI instruments deployed on Metop-A and Metop-B are presented and the relationships between NH3 amounts and temperature and precipitation are evaluated; the authors find a strong correlation between temperature and NH3 amounts in the Paris and Toronto domains, but only a weak one in the Mexico City domain. This analysis is extended to sub-domain scales, focusing on a number of previously identified source regions, and expanded to include the effect of relative humidity. Many of the smaller domains show an interesting local maxima in the NH3 vs temperature plot that coincides with fertilization activities. The impact of wind direction on air quality is also examined, using back-trajectories from HYSPLIT and a cluster analysis, along with in situ PM2.5 data from local networks. The PM 2.5 data is also used in conjunction with the NH3 columns to identify and count pollution events; wind roses are then constructed to determine the wind patterns on days with high pollution; interestingly PM2.5 and NH 3 are not always high on the same days. Finally IASI NH3 and the local PM2.5 data are compared against GEOS-Chem output for one month. This last analysis is interesting but too limited to provide really useful information on GEOS-Chem performance. The paper is well laid out and clearly written. The plots are of high quality and in general easily understood, though a few require more detailed captions, described below. It requires only some minor edits and clarifications to be acceptable for publication. Overall, a good illustration of how to apply a number of different techniques and data sources to the problem of understanding the drivers of pollution events.

Authors: We would like to thank the referee for this positive review and all the relevant comments that we have addressed in the following document.

**Technical comments**

Section 3.1: How are the IASI averaged and what is the grid resolution? How are the boxes doe each source region defined?

We have gridded the IASI data at 0.25° x 0.25° degrees. This information has been included in the revised manuscript. The sources regions are defined by looking at local enhancement in the 10-year average of the IASI total column.

Figure 3: Please comment on the high NH$_3$ values over the Arctic.

The sensitivity of IASI measurements is intimately related to the thermal contrast between the surface and the first layers of the atmosphere [Clerbaux et al., 2009]. When the detection is possible (with good thermal contrast), the peak sensitivity for NH$_3$ is in the boundary layer [Clarisse et al., 2010]. The high value over Canada and the Arctic in winter can be associated with high uncertainties

in the NH$_3$ retrievals due to low thermal contrast and high emissivity from snow. We have added in the revised manuscript this sentence to clarify: «The high value over Canada and the Arctic in winter can be associated with high uncertainties in the NH$_3$ retrievals due to low thermal contrast».

Lines 266-274: Is each back-trajectory associated with the 50 km NH3 mean for that day? Could the authors please briefly describe the clustering approach? Are the NH3 means clustered according to the corresponding back-trajectories?

We have explained the method following the different steps (Figure R1) and added these descriptions in the revised supplementary information Figure S7.

1) For each day, we have run HYSPLIT back-trajectories ending in the cities at the overpass time of the IASI satellite (blue lines in Figure R1).

2) For each day, we have calculated the amount of NH$_3$ derived from IASI observations within a circle of 50km radius around the cities (orange cylinder in Figure R1).

3) We have run the cluster analysis to merge trajectories that are near each other (green lines in Figure R1). The cluster analysis computes the spatial variance and minimize differences between trajectories within a cluster while differences between clusters are maximized [Abdalmogith et al., 2005; https://www.ready.noaa.gov/documents/Tutorial/html/traj_cluseqn.html]. NH$_3$ mean concentrations measured inside the cities by IASI have been allocated to the different mean cluster trajectories according to the corresponding back-trajectories.

[Figure]

*Figure R1: method to analyze the impact of long-range transport on NH$_3$ concentrations measured over the cities*

Line 289: Which sources have increased near Toronto?

According Yao and Zhang (2019), NH$_3$ concentrations around Toronto (near Granby – box A in Figure 2 and Elmira-Kitchener-Guelp – box B in Figure 2) have increased in the last decade. We have modified the sentence in the revised manuscript to clarify this point: "9 to 17% of NH$_3$ concentrations are coming from the west and the east of Toronto (cluster 3, 4, and 5) where atmospheric NH$_3$ have increased in the last decade (Boxes A and B in Figure 2, [Yao and Zhang, 2019])"

Line 303: What are the criteria used for defining a pollution event?

We have used the same method than the one conducted in Yamanouchi et al. (2021). Pollution events are defined when residuals of the Fourier fit are above 2 standard deviations. We have inserted "2 standard deviations" in the sentence of the revised manuscript.

Line 313: It would be useful to add the spring plot here

We have added the plot showing the seasonal occurrence of the pollution events in the revised manuscript.

Line 319: What does the radial distance in the wind roses in Figure 7 indicate?

The radial distance in the wind roses indicates the frequency of the wind direction occurrence. We have added this information in the revised manuscript.

Line 350: Please state the coincidence criteria.

The criteria were stated in the following sentence. To clarify, we have arranged this sentence: "Spatial and temporal coincidence criteria have been applied to GEOS-Chem outputs to compare with IASI morning observations, such as: model outputs between 8.30 and 11.30 AM coincident with IASI overpasses have been selected, and only collocated model outputs (at 0.5° × 0.625° horizontal resolution) have been selected coincident with IASI observations."

Lines 360-364: I don't agree that the GEOS-Chem and IASI are in good agreement; the authors should state that the two datasets capture some of the same pollution events.

We have modified the text accordingly.

Figure S4: Are the bins for the RH plot also specified by temperature? This does not seem right.

In Figure S4, lower panel, the $NH_3$ values are averaged per bins of relative humidity with a 1% RH between each consecutive bin. We modified the description of Figure S4 in order to make this clearer.

**Minor edits**

Authors: All minor edits have been addressed in the revised manuscript.

Lines 32-33: ... in Paris and Mexico City;

Line 65: However, NH3 concentrations are increasing in many countries: France, Canada and

Mexico reported increases of ......

Line 67: ... are composed of ...

Line 72: ...nitrate formed from ...

Line 127: ... are located within a 50-km

Line 134: ... the same month

Line 150: Note that

Line 153: ... daily 24-hour back-trajectories ...

Line 156: Finally, all back-trajectories are combined .... (not sure here what the authors mean to say)

We have modified this sentence as: "Finally, every back-trajectory that are near to each other are merged in clusters and associated with the corresponding local-scale IASI $NH_3$ concentrations".

Line 182: ... practices (which are dominant over Europe and North America)

Lines205: ... in this region closer to the Equator

Line 220: ...(with small contributions from industries)

Line 255: ...nitrogen fertilizers to

Line 267: concentrations. In order to analyze ...

Line 280: ... associated with the highest ...

Line 281: ...on average, are originating ...

Line 322: ... concentrations are observed

Line 352: ...numbers of coincident observations

Line 403: ...sources are

Line 440: ... surrounding regions:

Line 441: ...These lead

Line 444: ... and Mexico pollution is transported along the northeast-southwest line,

Line 458: ... launched in